# The subthalamic nucleus contributes causally to perceptual decision-making in monkeys

Kathryn Branam, Joshua I Gold, Long Ding*

Department of Neuroscience, University of Pennsylvania, Philadelphia, United States

**Abstract** The subthalamic nucleus (STN) plays critical roles in the motor and cognitive function of the basal ganglia (BG), but the exact nature of these roles is not fully understood, especially in the context of decision-making based on uncertain evidence. Guided by theoretical predictions of specific STN contributions, we used single-unit recording and electrical microstimulation in the STN of healthy monkeys to assess its causal, computational roles in visual-saccadic decisions based on noisy evidence. The recordings identified subpopulations of STN neurons with distinct task-related activity patterns that related to different theoretically predicted functions. Microstimulation caused changes in behavioral choices and response times that reflected multiple contributions to an 'accumulate-to-bound'-like decision process, including modulation of decision bounds and evidence accumulation, and to non-perceptual processes. These results provide new insights into the multiple ways that the STN can support higher brain function.

*For correspondence:
lding@pennmedicine.upenn.edu

## eLife assessment

The **fundamental** study by Ding and colleagues identifies subpopulations of neurons recorded in the monkey subthalamic nucleus (STN) with distinct activity profiles and causal contributions during perceptual decision-making. The combination of neuronal recording, microstimulation, and computational methods provides **convincing** evidence for a heterogenous neural population that could support multifaceted roles in decision formation. This study should be of wide interest to computational and experimental neuroscientists interested in cognitive function.

## Introduction

The subthalamic nucleus (STN) is a critical junction in both the indirect and hyperdirect pathways of the basal ganglia (BG). It receives inputs from the external segment of the globus pallidum (GPe) and cortex and sends diffuse excitation to pallidal output nuclei of the BG. The STN has well-recognized functions in movement control. For example, in humans and monkeys, lesions of the STN cause involuntary movements of contralateral body parts (*Martin, 1927*; *Martin and Alcock, 1934*; *Whittier and Mettler, 1949*; *Carpenter et al., 1950*). In monkeys with experimentally induced parkinsonism, STN lesions and inactivation can reverse abnormal BG output activity and alleviate both akinesia and rigidity (*Bergman et al., 1990*; *Bergman et al., 1994*; *Wichmann et al., 1994a*). In parkinsonian human patients, deep brain stimulation (DBS) of the STN has become a common treatment option to alleviate movement abnormalities (*DeLong and Wichmann, 2001*).

Recognizing that motor symptoms associated with STN damage are often accompanied by emotional and cognitive deficits, recent work has also begun to examine the roles of the STN in cognition. For example, the STN has been shown to contribute to cued, goal-driven action inhibition (*Baunez et al., 2001*; *Desbonnet et al., 2004*; *Witt et al., 2004*; *Aron and Poldrack, 2006*; *Frank*

**Figure 1.** Behavioral task and model predictions. (**A**) Behavioral task. The monkey was required to report the perceived motion direction of the random-dot stimulus by making a saccade toward the corresponding choice target at a self-determined time. (**B**) Three previous models predicted different patterns of subthalamic nucleus (STN) activity. Sensitive to choice: differential responses for trials ending with different choices. Sensitive to uncertainty: differential responses for trials with different evidence strength.

*et al., 2007*; *Isoda and Hikosaka, 2008*; *Schmidt et al., 2013*; *Pasquereau and Turner, 2017*). STN activity can also be sensitive to task complexity and decision conflict, as measured in imaging studies and human patients undergoing DBS (*Lehéricy et al., 2004*; *Aron et al., 2007*; *Fumagalli et al., 2011*; *Brittain et al., 2012*; *Zaghloul et al., 2012*; *Zavala et al., 2017*). These findings have led to the idea that STN may also contribute to resolving difficult decisions based on uncertain evidence. This idea has been formalized in several computational models, which posit three, not mutually exclusive, functions for STN: (1) through its interaction with GPe, STN computes a normalization signal to calibrate how the available, alternative options are assessed (*Bogacz and Gurney, 2007*; *Coulthard et al., 2012*; *Green et al., 2013*); (2) in coordination with the medial prefrontal cortex, STN adjusts decision bounds (i.e. thresholds on accumulated evidence that govern decision termination and commitment) to control impulsivity in responding (*Frank, 2006*; *Cavanagh et al., 2011*; *Ratcliff and Frank, 2012*; *Zavala et al., 2014*; *Herz et al., 2016*; *Herz et al., 2017*; *Pote et al., 2016*); and (3) by maintaining the balance between the direct and indirect pathways of the BG, STN helps to implement a nonlinear computation that improves the efficacy with which the BG adjusts decision bounds (*Lo and Wang, 2006*; *Wei et al., 2015*).

Guided by predictions of these models (*Figure 1B*), we assessed the role of the STN in decisions made by monkeys performing a random-dot visual motion direction discrimination task (*Figure 1A*). We recorded from individual STN neurons while monkeys performed the task and found activity patterns that were highly heterogenous across neurons. Nevertheless, these patterns could be sorted into three prominent clusters with functional properties that, in principle, could support each of the

three theoretically predicted STN functions from previous modeling studies. In addition, we tested STN's causal contribution to the decision process using electrical microstimulation. These perturbations of STN activity affected both choice and reaction time (RT) performance in multiple ways that could be ascribed to particular computational components of an 'accumulate-to-bound' decision process. As detailed below, these results show that STN can play multiple, causal roles in the formation of a deliberative perceptual decision, likely reflecting its diverse contributions to the many cognitive and motor functions that depend on the BG.

## Results

### STN neurons show diverse response profiles

We recorded 203 neurons while the monkeys were performing a random-dot motion discrimination task (*n*=115 and 88 for monkeys C and F, respectively). The behavioral performance of both monkeys has been documented extensively (*Ding and Gold, 2010*; *Ding and Gold, 2012a*; *Fan et al., 2018*). Their performance in three example sessions is shown in Figure 4A–C (black data points). In general, both monkeys made more contralateral choices with increasing signed motion strength (positive for motion toward the contralateral target; negative for motion toward the ipsilateral target) and had lower RTs (i.e. faster responses) for higher absolute motion strength.

STN neurons showed diverse response profiles. *Figure 2A* shows average activity patterns of three example neurons. The top neuron showed an initial suppression of activity after motion onset and became active, in a choice-dependent manner, before saccade onset. The middle neuron showed choice- and motion coherence-dependent activation during the motion-viewing period before saccade onset. The bottom neuron exhibited activation after motion onset that was similar for both choices and all coherence levels, which then decayed in a choice- and coherence-dependent manner around saccade onset.

The diversity of response profiles can be seen in the summary heatmaps for the population (*Figure 2B*). When activity was averaged across all trial types, STN neurons can become activated or suppressed (warm vs. cool colors, respectively), relative to pre-stimulus baseline, during motion viewing and around saccade onset. The timing of peak modulation also spanned the entire motion-viewing period and extended beyond saccade generation, including a substantial fraction of neurons that also responded to target onset before the motion stimulus appeared. These diverse spatiotemporal response profiles suggest that the STN as a whole may serve multiple functions in perceptual decision-making.

Across the population, a substantial fraction of neurons was sensitive to choice, motion coherence, and RT (*Figure 2C–E*, *Figure 2—figure supplement 1*). We performed multiple linear regressions, separately for coherence and RT (*Equations 1 and 2*), for each neuron and used the regression coefficients to measure these decision-related sensitivities. For choice sensitivity (*Figure 2C*, first row), both contralateral and ipsilateral preferences were commonly observed.

The overall fraction of neurons showing choice sensitivity increased after motion onset and peaked at saccade onset (*Figure 2D*). For coherence sensitivity, modulations were observed for trials with contralateral or ipsilateral choices and with similar tendencies for positive and negative coefficients (*Figure 2C*, rows 2 and 3). The fraction of neurons showing reliable coherence sensitivity was also higher around saccade onset (*Figure 2D*).

Despite the diverse distributions of regression coefficients, there were systematic patterns in when and how these forms of selectivity were evident in the neural responses. Notably, neurons showing choice sensitivity were more likely to show coherence modulation during early motion viewing, especially for trials when the monkey chose the neuron's preferred choice (*Figure 2E*, purple). In contrast, coherence modulation emerged later for neurons that did not show choice sensitivity (*Figure 2E*, gray lines). These systematic interactions in modulation types suggest that the STN population does not simply reflect a random mix of selectivity for decision-related quantities. Instead, there appears to exist subpopulations with distinct decision-related modulation patterns, which we detail below.

### STN subpopulations can support previously theorized functions

Using two forms of cluster analysis, we identified three subpopulations of neurons in the STN with distinct activity patterns that conform to predictions of each of the three previously published sets of

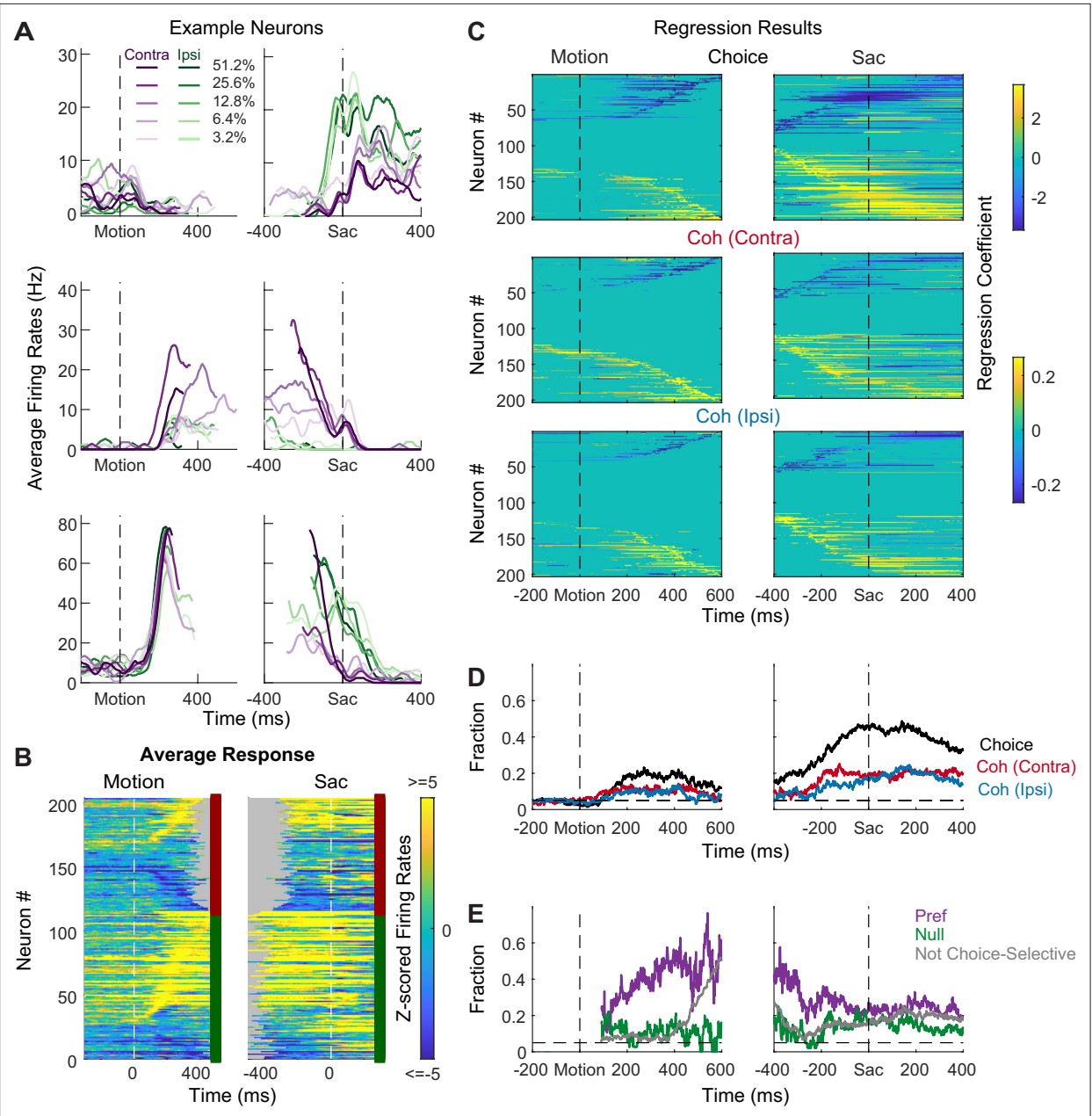

**Figure 2.** Subthalamic nucleus (STN) neurons have diverse response profiles. (**A**) Activity of three STN neurons (rows) aligned to motion (left) and saccade (right) onsets and grouped by choice × motion coherence (see legend). For motion-onset alignment, activity was truncated at 100 ms before saccade onset. For saccade-onset alignment, activity was truncated before 200 ms after motion onset. (**B**) Summary of average activity patterns. Each row represents the activity of a neuron, z-scored by baseline activity in a 300 ms window before target onset and averaged across all trial conditions. Rows are grouped by monkey (red and green shown to the right of each panel: monkeys F and C respectively) and sorted by the time of peak values relative to motion onset. Only correct trials were included. (**C**) Heatmaps of linear regression coefficients for choice (top), coherence for trials with contralateral choices (middle), and coherence for trials with ipsilateral choices (bottom), for activity aligned to motion (left) and saccade (right) onsets. Regression was performed in running windows of 300 ms. Regression coefficients that were not significantly different from zero (t-test, p>0.05) were set to zero (green) for display purposes. Neurons were sorted in rows by the time of peak coefficient magnitude. Only correct trials were included. (**D**) Time courses of the fractions of regression coefficients that were significantly different from zero (t-test, p<0.05), for choice (black), coherence for trials with contralateral choices (red), and coherence for trials with ipsilateral choices (blue). Dashed line indicates chance level. (**E**) Time courses of the fractions of non-zero regression coefficients for coherence. Separate fractions were calculated for trials with the preferred (purple) and null (green) choices from choice-selective activity and for all trials from activity that was not choice selective (gray). Only time points after motion onset with fractions > 0.05 for choice-selective activity were included. Dashed line indicates chance level.

The online version of this article includes the following figure supplement(s) for figure 2:

**Figure supplement 1.** Subthalamic nucleus (STN) activity is modulated by choice and reaction time (RT).

models. For the first analysis, we represented a neuron's activity pattern with a 30-dimensional (30D) vector, consisting of normalized average activity associated with two choices, five coherence levels, and three task epochs. We generated three artificial vectors based on the predicted activity patterns of each model, as follows (*Figure 3A*). *Bogacz and Gurney, 2007*, posited that STN neurons, through their reciprocal connections with the external segment of GPe, pool and normalize evidence-related signals, leading to the prediction of choice and coherence-modulated activity during motion viewing (*Figure 3A*, top; based on simulations using equations in their Appendix B). *Ratcliff and Frank, 2012*, posited that STN neurons, through their direct innervation by cortical regions, provide an early signal to suppress immature choices, leading to the prediction of a choice-independent signal that appears soon after motion onset and dissipates over time (*Figure 3A*, middle; based on their Figure 5). *Wei et al., 2015*, posited that the STN balances evidence-related signals in the GPe until near decision time, leading to the prediction of coherence-dependent ramping activity with no or weak choice selectivity (*Figure 3A*, bottom; based on their *Figure 2D*). We performed *k*-means clustering using these three vectors and another arbitrary vector as the seeds to group the population into four clusters.

*Figure 3B* shows the average activity from each of the resulted clusters. Consistent with the design of this analysis, the first cluster tends to show choice- and coherence-dependent activity that also ramps up during motion viewing (*Figure 3B*, first row; *Figure 3—figure supplement 1*). The second cluster tends to show an early, sharper rise in activity during motion viewing and this activity gradually decreases toward saccade onset (*Figure 3B*, second row). The third cluster tends to show ramping activity during motion viewing with similar coherence modulation for both choices and a short burst of activity for one choice just before saccade onset (*Figure 3B*, third row; *Figure 3—figure supplement 1*). The last cluster shows mixed and, on average, weak task-related modulation (*Figure 3B*, bottom row; *Figure 3—figure supplement 1*). The first three clusters contained similar numbers of neurons. When visualized using the *t*-distributed stochastic neighbor embedding (*t*-SNE) technique, these clusters did not form a single continuum but instead reflected separable features between clusters (*Figure 3C*). In other words, the clustering did not simply force a uniform distribution with random-mixed selectivity into four groups.

For the second cluster analysis, we used random seeds without considering any of the model predictions and obtained almost identical clusters. As detailed in Methods, we explored a wide range of settings for clustering, including: (1) using directly the 30D vectors or their principal component projections, (2) basing the clustering on three different distance metrics, and (3) varying the number of presumed clusters. To identify the best setting, we assessed the goodness of clustering using the silhouette score and the stability of clustering using the Rand index (*Rand, 1971*; *Figure 4*). The silhouette score quantifies for each member the relative distance between its average within-cluster distance and distance to those in its closest neighboring cluster (a higher score indicates better cluster separation). The silhouette plots favored the combination of using the 30D vector directly and correlation distance (*Figure 4A*, third row), which generated less variability across clusters and few small-magnitude negative silhouette scores (negative scores indicate that a member is closer to its neighboring cluster than its own cluster).

The Rand index measures how consistently two members are assigned to the same clusters from different iterations of clustering (a high index indicates greater stability). The Rand index was generally high (above 0.95 out of a max of 1; blue box) except for the combination of using the 30D vector and cosine distance (*Figure 4B*). Finally, using the raw vector-correlation combination, an assumption of four clusters resulted in the highest average Rand index and assuming four to six clusters generally resulted in higher mean silhouette score and lower number of negative scores (*Figure 4C*; blue arrow). We thus considered that the raw vector-correlation combination and an assumption of four clusters produced the most stable and plausible results.

As shown in *Figure 3D and E*, the four clusters thus identified closely matched those obtained using model predicted seeds, in terms of the average activity, the cluster sizes, and their locations in the *t*-distributed stochastic neighbor embedding (*t*-SNE) space. Increasing the assumed number of clusters caused changes mostly in the gray cluster, with some changes in the blue cluster, and little effects on the red and purple clusters (*Figure 4—figure supplements 1 and 2*). Together these results suggest that, absent the ground truth on the number of subpopulations in STN, there exist at least three subpopulations that each corresponds to the predictions of one of three previously published

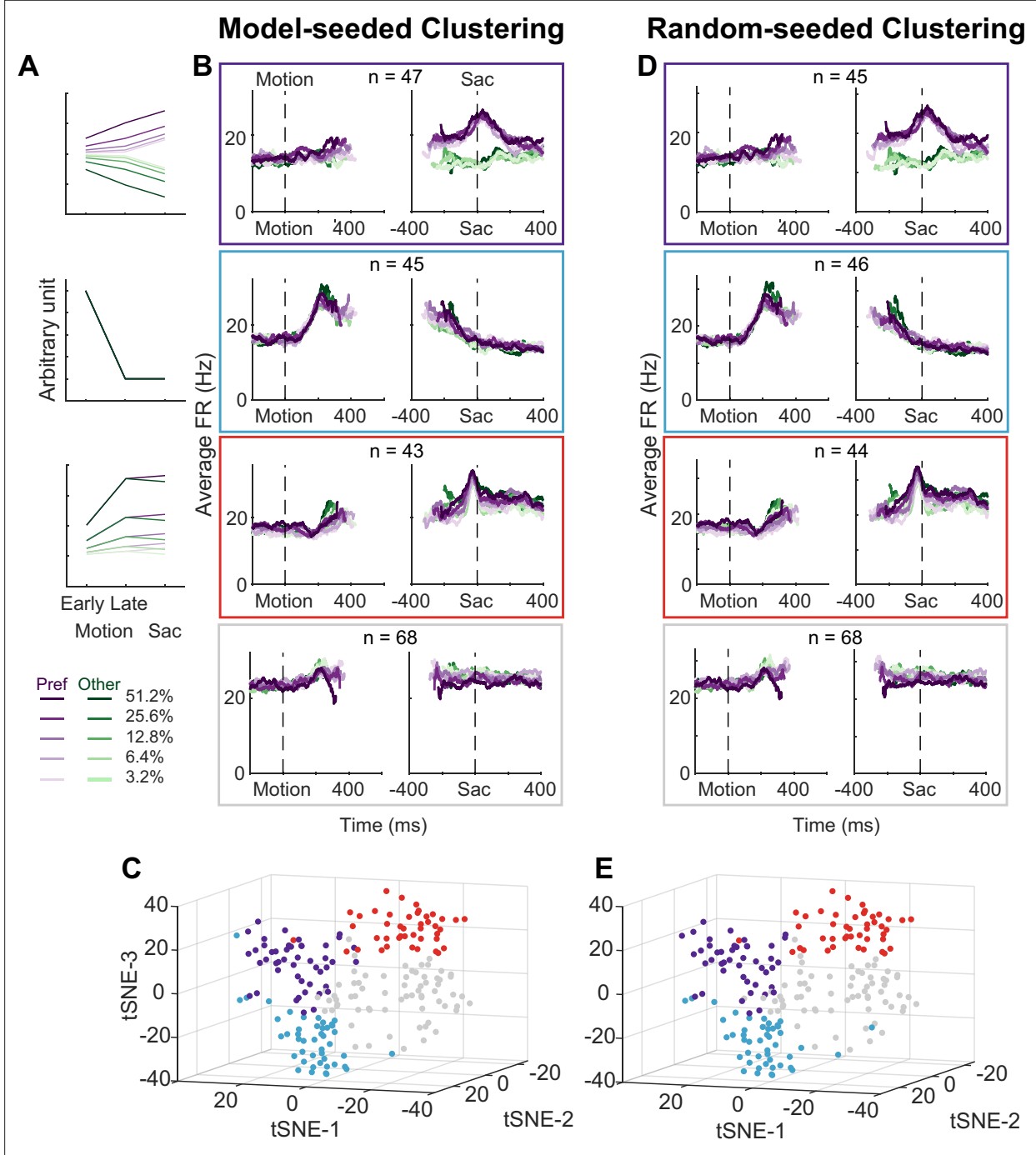

**Figure 3.** Subthalamic nucleus (STN) contains distinct subpopulations. (**A**) Three activity vectors that were constructed based on theoretical predictions in *Figure 1B* and used as seeds for *k*-means clustering (see Methods). (**B**) Each panel shows the average activity of neurons in a cluster, same format as *Figure 2A*. The numbers indicate the cluster size. (**C**) Visualization of the clusters using the *t*-distributed stochastic neighbor embedding (*t*-SNE) dimension-reduction method. (**D**) Average activity of clusters identified using random-seeded *k*-means clustering. Same format as Figure 3B. (**E**) Visualization of the random-seeded clusters in the same *t*-SNE space.

The online version of this article includes the following figure supplement(s) for figure 3:

**Figure supplement 1.** Summary of regression results, separated for different subpopulations.

**Figure supplement 2.** Indices of motivational state did not differ among sessions with different neuron subpopulations.

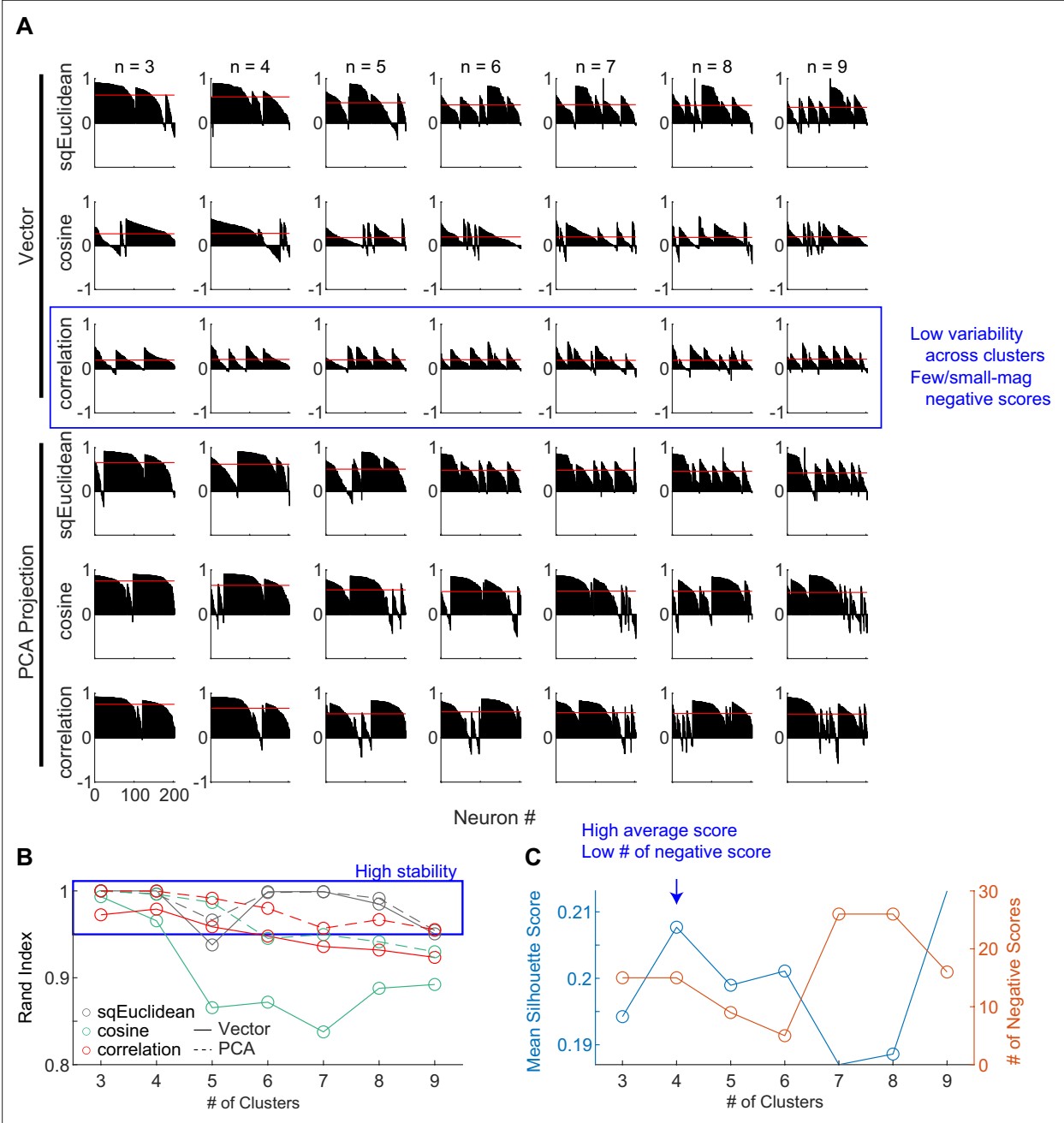

**Figure 4.** Clustering parameters. (**A**) Silhouette plots for clustering results using different combinations of settings. Silhouette scores for neurons are grouped by clusters and sorted. Red lines indicate the mean scores. (**B**) Average Rand indices for different clustering settings. For each setting, the *k*-means algorithm was run 50 times, each time picking the best clusters out of 100 repetitions. Higher Rand index indicates greater cluster stability across different runs. Blue box indicates settings with Rand indices > 0.95. (**C**) Mean silhouette scores and the number of negative scores as a function of number of clusters, using the firing rate vectors and correlation distance. Higher mean score and fewer negative scores indicate better clustering.

The online version of this article includes the following figure supplement(s) for figure 4:

**Figure supplement 1.** Clustering results using alternative numbers of clusters, visualized in *t*-distributed stochastic neighbor embedding (*t*-SNE) space.

**Figure supplement 2.** Clustering results using alternative numbers of clusters, visualized as average firing rates for each cluster.

models. As a consequence, STN appears in principle to be able to support multiple decision-related functions.

## Perturbation of STN activity affects choice and RT

To better understand STN's functional roles in the decision process, we perturbed STN activity using electrical microstimulation while monkeys performed the task. Specifically, we applied a train of current pulses at identified STN sites during decision formation, lasting from motion onset to saccade onset. *Figure 5* shows microstimulation effects on choices and RTs in three example sessions. In the first example session, STN microstimulation caused a leftward horizontal shift (more contralateral choices) and slope reduction (more variable choices) in the psychometric curve (*Figure 5A*, top), as well as a substantial flattening of RT curves (faster responses that depended less on motion coherence) for both choices (bottom). In the second example session, STN microstimulation induced a minor leftward shift in the psychometric curve and asymmetric changes in RT for the two choices (*Figure 5B*). In the third example session, STN microstimulation did not change the psychometric curve but caused reductions in RT for both choices (*Figure 5C*).

Across 54 different STN sites, microstimulation caused variable choice biases and tended to reduce the dependence of choice on motion strength. We fitted a logistic function to the choice data and measured choice bias (horizontal shift) and motion strength dependence (slope). In 23 sessions, microstimulation induced a reliable choice bias (*Figure 5D*). The induced bias was toward the contra-lateral or ipsilateral choice in 15 and 8 sessions, respectively, and the median value for bias was not significantly different from zero (Wilcoxon signed-rank test, p=0.15). In 18 sessions, microstim-ulation induced a change in the slope. The slope was reduced in 15 sessions and the median value was negative (p=0.008). These tendencies were robust across different variants of logistic functions, with or without lapse terms to capture errors independent of motion strength (*Figure 5—figure supplement 1*). Of the sessions where inclusion of lapse terms for the control and microstimulation trials produced lower Akaike information criteria (AICs), very few showed significant microstimulation-induced changes in lapses (2 sessions each for the 'Symmetric Lapse' and 'Asymmetric Lapse' vari-ants). Thus, based on fitting results using logistic functions, STN microstimulation most consistently reduced the choice dependence on motion strength, caused session-specific choice biases, and had minimal effects on lapses.

Microstimulation also tended to reduce RT. We fitted linear functions separately for RTs associated with the two choices, in which offset and slope terms measure coherence-independent and -depen-dent changes in RTs, respectively. Microstimulation caused changes in RT offsets in 25 sessions for contralateral choices (18 were reductions in RT, with a median change across all sessions of –36 ms; Wilcoxon signed-rank test for $H_0$: median change = 0, p<0.0001) and 18 sessions for ipsilateral choices (15 reductions, mean change = –23 ms, p<0.0001; *Figure 5E*). Microstimulation caused changes in RT slopes in 6 sessions for contralateral choices (5 were positive, implying a weaker coherence depen-dence; p<0.0001) and 4 sessions for ipsilateral choices (all 4 were positive; *Figure 5F*). Thus, based on fitting results using linear functions, STN microstimulation can induce choice-specific changes in RT, with overall tendencies to reduce both the coherence-independent component and the RT's depen-dence on coherence for the contralateral choice.

## Microstimulation effects reflected changes in multiple computational components

To infer STN's computational roles in the decision process, we examined the microstimulation effects using a drift-diffusion model (DDM) framework. This framework has been widely used in studies of perceptual decision-making and can provide a unified, computational account of both choice and RT (*Gold and Shadlen, 2007*). It assumes that noisy evidence is accumulated over time and a decision is made when the accumulated evidence reaches a certain decision bound. The overall RT is the sum of the time needed to reach the bound and non-decision times reflecting perceptual and motor laten-cies. Previous theoretical models also made predictions about the effects of perturbing STN activity that can be interpreted in the DDM framework. The model by *Bogacz and Gurney, 2007*, predicted that the perturbation would reduce the effect of task difficulty on decision performance by eliminating a nonlinear transformation that is needed for appropriate evidence accumulation (*Green et al., 2013*). The model by *Ratcliff and Frank, 2012*, predicted that the perturbation, by causing changes in the

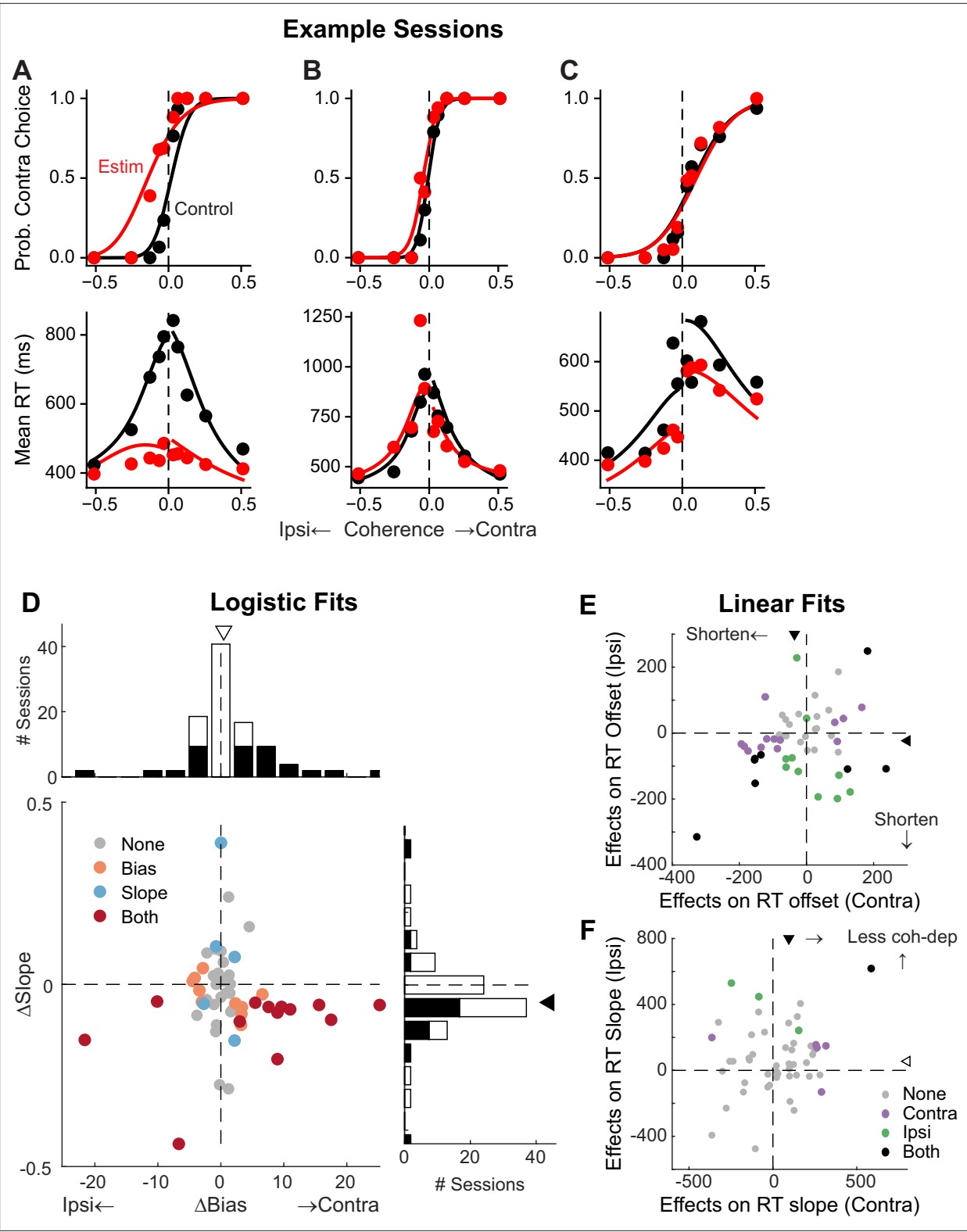

**Figure 5.** Subthalamic nucleus (STN) microstimulation affects monkeys' choice and reaction time (RT). (**A–C**) Monkey's choice (top) and RT (bottom) performance for trials with (red) and without (black) microstimulation for three example sessions (**A**,**B**: two sites in monkey **C**; **C**: monkey **F**). Lines: drift-diffusion model (DDM) fits. (**D**) Distributions of microstimulation effects on bias and slope terms of the logistic function. Filled bars in histograms indicate sessions with significant modulation of the specific term (bootstrap method). Triangles indicate the median values. Filled triangle: Wilcoxon signed-rank test for $H_0$: median = 0, p<0.05. (**E** and **F**) Summary of microstimulation effects on the offset (**E**) and slope (**F**) terms of a linear regression fit

*Figure 5 continued on next page*

*Figure 5 continued*

to RT data. Two separate linear regressions were performed for the two choices (Ipsi/Contra, as indicated). Triangles indicate the median values. Filled triangles: Wilcoxon signed-rank test, p<0.05.

The online version of this article includes the following figure supplement(s) for figure 5:

**Figure supplement 1.** Comparison of different logistic models.

STN's influence onto the substantia nigra pars reticulata, would change temporal dynamics of the decision bound and influence non-decision time. The model by *Wei et al., 2015*, predicted that the perturbation would result in a reduction in the decision bound.

To test whether these predictions, and/or other effects, were present in our microstimulation data, we fitted a DDM to choice and RT data simultaneously (*Figure 6A*). We performed AIC-based model selection and found that, in 40 of 54 sessions, the Full model, which included microstimulation effects on any model parameters, outperformed the None model, which assumed that there was no microstimulation effect on any parameters (*Figure 6B*). This result implies that, in these sessions, STN microstimulation affected one or more computational components of the decision process. To better characterize these effects, we compared AICs between the Full model and six reduced models to identify sessions with reliable microstimulation-induced changes in particular model parameters (*Figure 6—figure supplement 1A*).

We found that STN microstimulation resulted in reliable changes in several model parameters over different subsets of sessions (*Figure 6C and D*). Consistent with model predictions from *Bogacz and Gurney, 2007*, microstimulation reduced the scale factor for evidence accumulation, $k$, in 14 sessions. This effect contributed to a decreased motion coherence dependence of choice and RT (*Figure 6C*, first histogram; Wilcoxon signed-rank test for $H_0$: zero median effect, p=0.021). Consistent with model predictions from *Ratcliff and Frank, 2012*, and *Wei et al., 2015*, microstimulation affected parameters that controlled the decision bound ($a$, $B\_collapse$, $B\_t$) in 16 sessions each (not necessarily in the same sessions for each parameter, see *Figure 6D*). The changes in the maximal decision bound ($a$) were variable across sessions (p=0.68). The changes in the collapsing bound dynamics ($B\_collapse$, $B\_t$) tended to indicate faster and earlier decreases in bounds (p=0.039 and 0.088, respectively). Consistent with model predictions from *Ratcliff and Frank, 2012*, microstimulation caused changes in non-decision times in 30 sessions ($t0\_Contra$ and $t0\_Ipsi$). These changes varied from session to session (p=0.28 and 0.75, respectively). Statistical tests on fitted parameters of all sessions, regardless of whether a microstimulation effect was necessary to account for the behavioral data, showed similar trends (*Figure 6—figure supplement 1B*).

STN microstimulation had two additional effects beyond those predicted by the previous modeling studies. First, consistent with the above-demonstrated microstimulation-induced choice biases (*Figure 5*), microstimulation induced offsets in momentary ($me$; n=16 sessions; p=0.61) and accumulated ($z$; n=12 sessions; p=0.016) evidence. Second, the microstimulation effects involved changes in more than one model parameter in the majority of sessions (*Figure 6D*). We did not observe any dominant combinations of effects. These results suggest that the STN is causally involved in multiple decision-related functions, including those mediating the dependence on evidence, choice biases, and bound dynamics.

## Distribution of microstimulation effects reflected intermingled neuron activity patterns

The multi-faceted microstimulation effects, combined with the fact that the kind of microstimulation we used tends to activate not just one neuron, but rather groups of neurons near the tip of the electrode (*Tehovnik, 1996*), suggested that STN neurons with different functional roles are located close to one another. Consistent with this idea, neurons that were classified as belonging to different clusters tended to be intermingled (*Figure 7*). We did not observe any consistent topographical organization patterns within or between the two monkeys. At certain locations, neurons belonging to different clusters were recorded using the same electrode. We calculated silhouette scores to quantify whether the activity pattern-based neuron clusters also formed clusters in the 3D physical space. The mean values were –0.09 and –0.11 for the two monkeys, respectively, indicating that neurons were often closer to others from a different cluster than those within the same cluster. In other words, STN

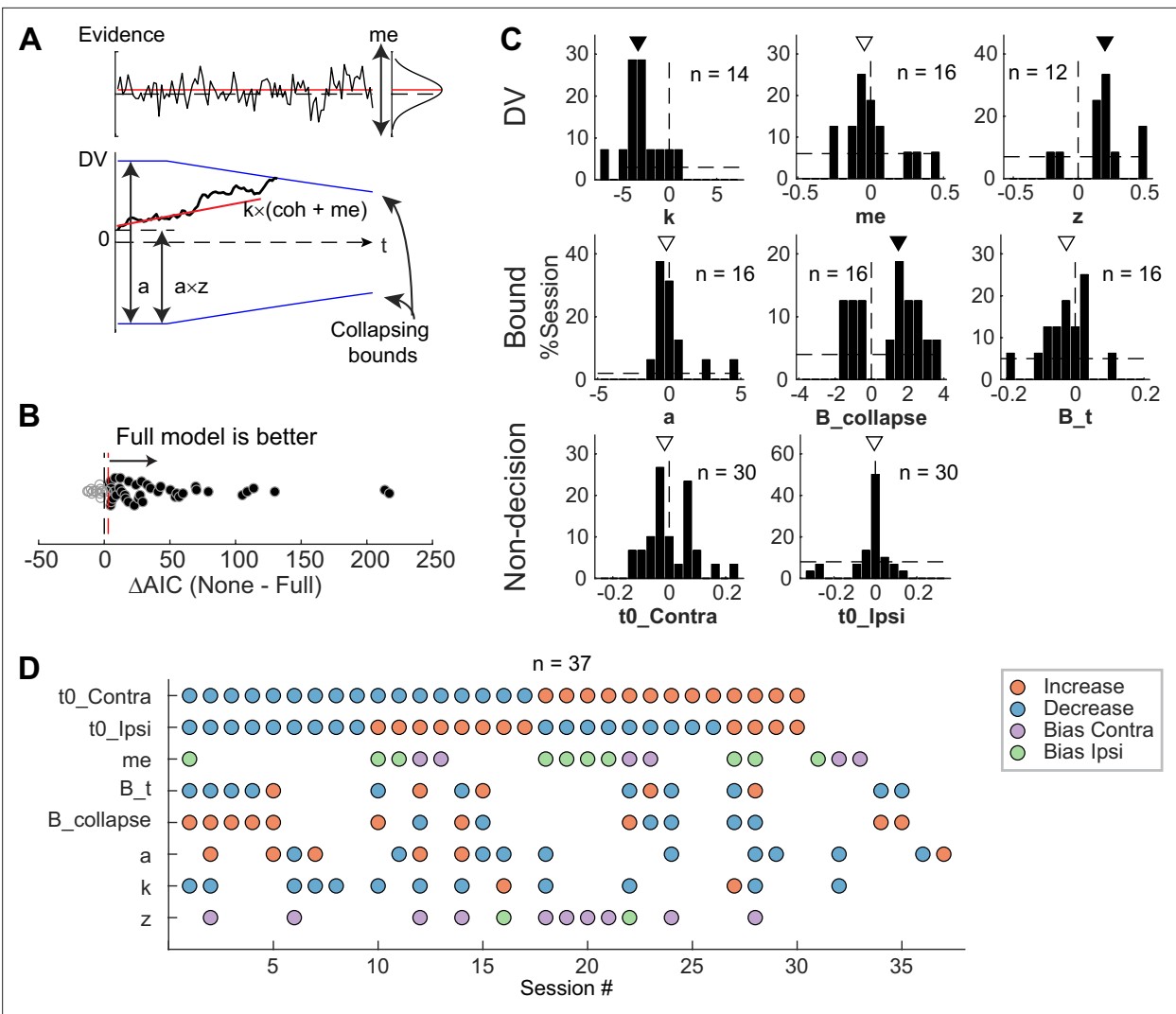

**Figure 6.** Subthalamic nucleus (STN) microstimulation affected multiple computational components in the drift-diffusion model (DDM). (**A**) Illustration of the DDM. Red/black lines represent across-trial mean/single-trial example of the evidence (top) and drift rate (bottom). Blue lines represent the collapsing decision bounds. (**B**) Distribution of the difference in Akaike information criterion (AIC) between the None and Full models. Red dashed line indicates the criterion for choosing the Full model: AIC difference = 3. (**C**) Histograms of microstimulation effects on DDM parameters. Each histogram included only sessions in which the Full model outperformed the corresponding reduced model (e.g. the histogram for parameter *a* included only sessions in which AIC$_{NoA}$ – AIC$_{Full}$ >3 and AIC$_{None}$ – AIC$_{Full}$ >3). Triangles indicate median values. Filled triangles: Wilcoxon signed-rank test, p<0.05. (**D**) Summary of microstimulation effects on all parameters, for sessions in which at least one significant effect was present. Sessions were sorted by the prevalence and sign of the effects.

The online version of this article includes the following figure supplement(s) for figure 6:

**Figure supplement 1.** Supplemental DDM fitting results.

(**A**) Differences in Akaike information criterion (AIC) between reduced and Full models. Filled circles indicate sessions for which AIC$_{Reduced}$ – AIC$_{Full}$ >3 (red line). Note that for three sessions, the Full model outperformed the None model but not any of the reduced models. (**B**) Histograms of difference in drift-diffusion model (DDM) parameters between trials with and without microstimulation. Filled bars represent sessions considered to show significant microstimulation effects on the given parameter, based on AIC comparisons. Triangles indicate median values. Filled triangles: Wilcoxon signed-rank test, p<0.05.

subpopulations did not segregate from each other and instead tended to be intermingled, and thus microstimulation likely activated multiple neurons with different functional properties.

Although the intermingled organization of STN subpopulations, defined based on their task-related activity patterns (*Figure 3*), made it challenging to relate a specific microstimulation effect to a specific subpopulation, we did observe certain trends that could contribute to the site-specific

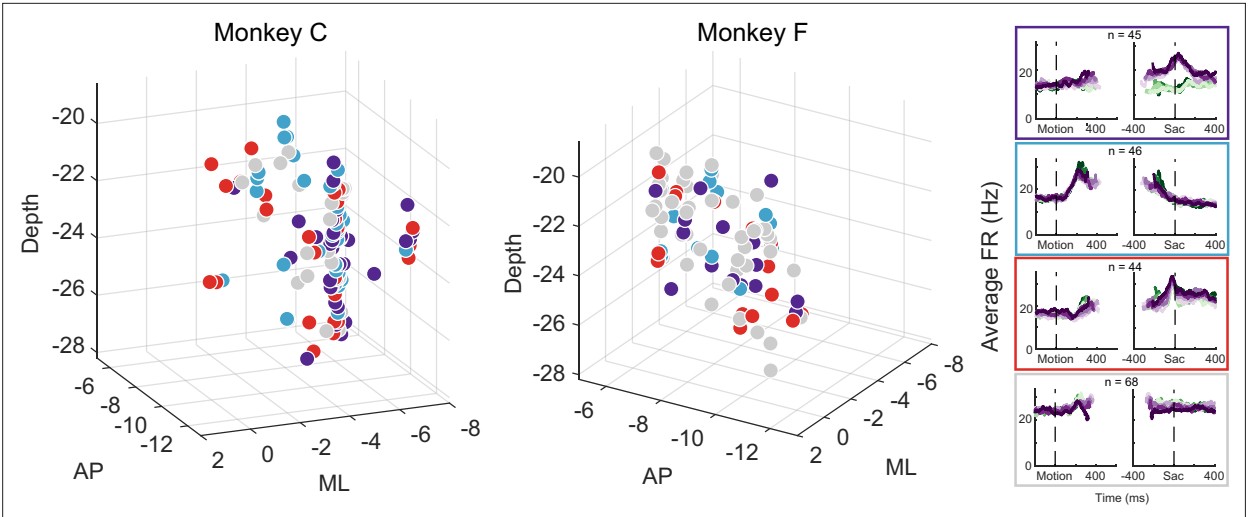

**Figure 7.** Different subthalamic nucleus (STN) subpopulations are intermingled. Locations of STN neurons, color-coded by clusters based on random-seed clustering (same as *Figure 3D*). The medial-lateral (ML) values were jittered for better visualization of neurons recorded along the same track and at similar depths. Anterior-posterior (AP) levels were relative to the anterior commissure. ML and depth levels were relative to the center of the recording chambers.

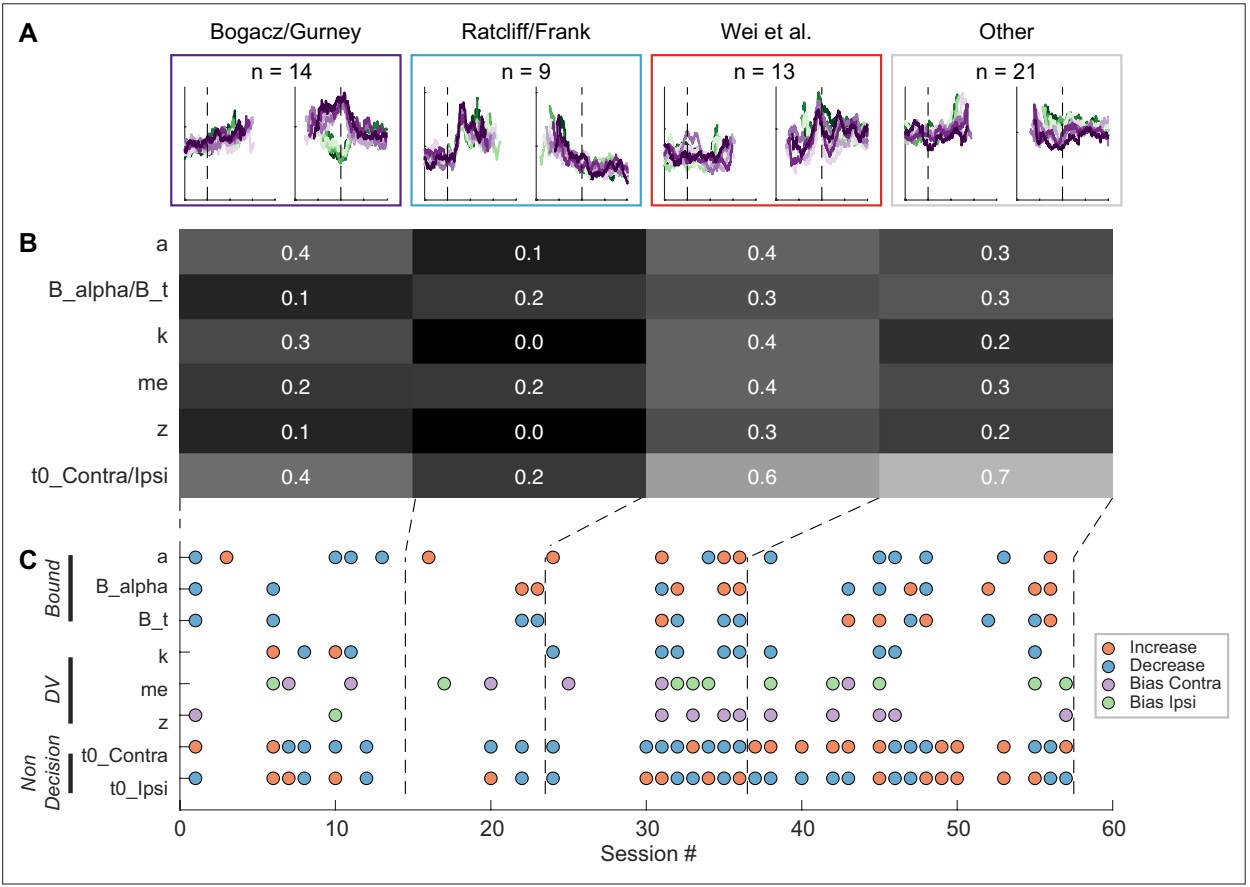

**Figure 8.** Subthalamic nucleus (STN) microstimulation effects depend partially on neural clusters. (**A**) Average activity at stimulation sites, grouped by four clusters based on the clusters in *Figure 3D*. (**B**) Fractions of significant microstimulation effects for sites with the presence of each neuron cluster. Significance was based on Akaike information criterion (AIC) comparison between reduced and Full models. (**C**) Microstimulation effects grouped by neuron cluster. Same format as *Figure 6D*.

microstimulation effects. We assigned the single- or multi-unit activity at the stimulation sites to the clusters identified using the random-seeded clustering (*Figure 8A*). We then grouped the sites by neuron clusters (*Figure 8C*). When neurons of different clusters were recorded at the same site, the same microstimulation effects were assigned to each cluster. We found that the second cluster was associated with lower overall likelihood of observing microstimulation effects compared to other clusters (*Figure 8B*; Chi-square test, $H_0$: the likelihood is the same for the first cluster and the other clusters; p=0.003), while the third cluster had higher overall likelihood (p=0.035). For the first three clusters, no microstimulation effect dominated (p>0.3 for all), whereas for the fourth cluster, it was more likely to observe effects on non-decision times (i.e. with neural activity patterns not related to the three models; p=0.001).

The sign of microstimulation effects depended weakly on neuron clusters. For example, it was more likely to observe an increase in maximal bound height ('*a*') for the third neuron cluster (Chi-square test, $H_0$: same fractions of increase/decrease for all clusters; p=0.073; Chi-square test, $H_0$: equal fractions of increase/decrease within the cluster; p=0.036). Microstimulation decreased the scale factor ('*k*') for the third and fourth clusters but caused variable changes for the first cluster (p=0.070 and 0.021, respectively). Microstimulation effects on the non-decision time for the contralateral choices were dominated by increases for the fourth cluster (p=0.04 and 0.007, respectively). Together, these results suggested that microstimulation effects reflected multiple contributions of intermingled STN subpopulations to decision- and non-decision-related processes.

## Heterogeneous activity patterns and microstimulation effects cannot be explained by variations in motivational state

Another potential source of heterogeneity in our data may reflect variations in the monkeys' motivational state across sessions. In two sets of analyses, we did not observe any significant influence of motivational state on the recording or microstimulation data. For these analyses, we used the rate of fixation breaks, overall error rate, and mean RT as indices of motivational state. None of these measurements differed among sessions when different subpopulations were encountered (*Figure 3—figure supplement 2*), suggesting that the motivational state cannot predict which type of activity pattern would be observed. Similarly, none of these measurements significantly correlated with the microstimulation effects in any DDM component (*Supplementary file 1*), suggesting that the motivational state did not modulate the magnitude of microstimulation effects. Together, these results suggest that the diverse activity patterns and microstimulation effects cannot be accounted for by variations in the monkeys' task engagement.

## Discussion

We provide the first characterization of single-unit recordings and electrical microstimulation in the STN of monkeys performing a demanding perceptual-decision task. We show that: (1) STN neurons are heterogeneous in their response profiles; (2) different STN subpopulations, with distinct decision-related activity modulation patterns and intermingled within the region, can support previously theorized functions; and (3) electrical microstimulation in STN causes changes in choice and RT behaviors, reflecting effects on multiple computational components of an accumulate-to-bound decision process. These results indicate that the STN plays important and complex roles in perceptual decision formation, both supporting and extending existing views of STN function.

Our study was motivated by the differing predictions of STN activity patterns from several theoretical studies that were based on STN cellular physiology, connectivity, and/or response patterns in non-perceptual decision-making contexts (*Bogacz and Gurney, 2007*; *Ratcliff and Frank, 2012*; *Wei et al., 2015*). Remarkably, we found three clusters of STN activity that are consistent with each of these predictions. The three clusters were robust and stable, emerging when we used two different clustering methods (one with model-based seeds, the other with random seeds). Interestingly, *Zavala et al., 2017*, have reported two types of STN responses in human patients performing a flanker task. The 'early' response they identified may correspond to our second cluster, whereas the 'late' response may reflect a combination of our first and third clusters. Together these results suggest that the primate STN contains distinct subpopulations with different functional roles. Combined with the microstimulation results, the presence of these subpopulations suggests that the STN can both

contribute to the conversion of sensory evidence into an appropriately formatted/calibrated decision variable (DV) and modulate the dynamics of decision bound. Future studies of BG function should strive to better understand how these subpopulations interact with each other, as well as with other neurons in the BG and the larger decision network to support decision making.

Despite the general agreements between our observations and previous theoretical predictions, there were also differences that could be informative for developing future BG models. Most notably, previous models focused on the period of evidence accumulation and less on neural activity patterns at or after decision commitment. In contrast, our data show interesting modulations around saccade onset (see *Figure 3*) that raise several intriguing possibilities for STN's contributions to the decision process. In particular, one subpopulation showed a broad peak with strong choice modulation and little coherence modulation. These modulations may reflect bound-crossing in an accumulate-to-bound process (but see below). A second subpopulation returned to the baseline level before saccade onset. The relatively constant trajectory of this modulation may reflect a collapsing bound or urgency signal that is dependent only on elapsed time and not the sensory evidence. A third subpopulation maintained coherence-dependent activity until very close to saccade onset and showed a sharp peak with little choice or coherence modulation. This sharp peak may signal the end of decision deliberation, without specifying which decision is made, to direct the network to a post-decision state for decision evaluation.

The diverse activity patterns and their intermingled distribution in the STN underscore the challenge of identifying specific, causal contributions of a particular neural subpopulation. In many sessions, we observed effects that have been predicted theoretically and observed experimentally in human PD patients undergoing DBS. These effects included a reduction in RT, a weaker dependence on evidence, and changes in the maximal value and trajectories of the decision bound (*Frank et al., 2007*; *Cavanagh et al., 2011*; *Coulthard et al., 2012*; *Green et al., 2013*; *Zavala et al., 2014*; *Herz et al., 2016*; *Pote et al., 2016*). In addition to these previously observed effects, microstimulation also changed choice biases, measured as horizontal shifts of psychometric functions and as two different types of biases in the DDM framework. This departure from previous DBS studies may arise from different task designs (button press versus eye movement), health status of the subjects, and experience level (minimally versus extensively trained). The lateralized bias suggests that the STN may be involved in flexible decision processes that adapt to environments with asymmetric prior probability and/or reward outcomes for different alternatives, in addition to modulating speed-accuracy tradeoff. Consistent with this idea, DBS can affect the threshold for deliberations over uncertain sensory inputs or motivational factors such as reward and effort (*Pagnier et al., 2024*), suggesting that the STN may be part of a general selection machinery that can incorporate sensory evidence with information about the task environment (*Redgrave et al., 1999*).

The intermingled subpopulations may appear at odds with the conventional idea of topography in how the STN is organized. For example, the 'tripartite model' suggests that STN is segregated by motor, associative, and limbic functions (*Parent and Hazrati, 1995*); afferents from motor cortices and neurons related to different types of movements are largely somatotopically organized in the STN (*DeLong et al., 1985*; *Nambu et al., 1996*); and certain molecular markers are expressed in an orderly pattern in the STN (reviewed in *Prasad and Wallén-Mackenzie, 2024*). Because we focused on STN neurons that were responsive on a single oculomotor decision task, our sampling was likely biased toward STN subdivisions related to associative function and oculomotor movements. As such, our results do not preclude the presence of topography at a larger scale. Rather, our results underscore the importance of activity pattern-based analysis, in addition to anatomy-based analysis, for understanding the functional organization of the STN.

Our findings also suggest that STN's role in decision formation differs in important ways from other oculomotor regions that have been examined under similar conditions. First, in the frontal eye field (FEF), lateral intraparietal area (LIP), and superior colliculus (SC), decision-related neural activity is dominated by a choice- and coherence-dependent 'ramp-to-bound' pattern (*Roitman and Shadlen, 2002*; *Ding and Gold, 2012a*; *Crapse et al., 2018*; *Cho et al., 2021*; *Jun et al., 2021*; *Stine et al., 2023*), with additional multiplexing of decision-irrelevant signals (*Meister et al., 2013*). In contrast, different STN subpopulations can carry distinct signals that may all be relevant to decision formation. Moreover, these signals include patterns not evident in the other regions, such as a choice- and

coherence-independent activation in early motion viewing (blue cluster in *Figure 3B and D*), that may signal a unique role for the STN.

Second, choice-selective ramping activity has been identified in LIP, FEF, SC, the caudate nucleus, and now two STN subpopulations (*Ding and Gold, 2010*; *Fan et al., 2020*). However, such activity differs among these oculomotor regions just before saccade onset for the preferred choice. In LIP, FEF, and SC, when the ramping activity is aligned to saccade onset, it shows negative coherence modulation and positive RT modulation before converging to a common, higher level, consistent with an accumulate-to-bound process. In the caudate nucleus, the ramping activity does not converge to a common, higher level. For the first STN subpopulation (*Figure 3*), the ramping activity showed on average positive coherence modulation and negative RT modulation (opposite to predictions of an accumulate-to-bound process) before converging to a common, higher level. The second STN subpopulation did not show choice-selective activity before saccade onset. These differences suggest that the caudate and STN neurons participate in decision deliberation but do not directly mediate decision termination (bound crossing). It is also possible that the ramping activity reflects some roles for the STN in the evaluation of the decision process, the tracking of elapsed time, or both. How these possible roles relate to those of caudate neurons awaits further investigation (*Fan et al., 2024*).

Third, whereas unilateral perturbations in LIP and SC tend to induce contralateral choice biases (*Hanks et al., 2006*; *Jun et al., 2021*; *Jeurissen et al., 2022*; *Stine et al., 2023*), unilateral STN (and caudate) microstimulation can induce both contralateral and ipsilateral choice biases, depending on the stimulation site (*Ding and Gold, 2012b*; *Doi et al., 2020*). At many sites, STN microstimulation effects on RT were often bilateral and of the same polarity. Moreover, STN microstimulation seems to have a particularly strong effect on the overall dependence of choice and RT on evidence, which was not the case for other oculomotor regions. These differences suggest that the STN has unique roles in choice-independent computations, potentially including those involving evidence pooled for all alternatives or general bound dynamics (*Bogacz and Gurney, 2007*; *Ratcliff and Frank, 2012*).

In summary, we characterized single-neuron activity and the effects of local perturbations in the STN of monkeys performing a deliberative visual-oculomotor decision task. Our results validated key aspects of previous theoretical predictions, providing experimental evidence for the multiple involvement in modulating decision deliberation and commitment. Our results also identified other features of decision-related processing in STN that differ from both theoretical predictions and known properties of other brain areas that contribute to these kinds of decisions. These differences can help guide future investigations that aim to delineate how cortical-subcortical interactions in general, and interactions involving the STN in particular, support decision-making and other aspects of higher brain function.

# Methods

**Key resources table**

| Reagent type (species) or resource | Designation | Source or reference | Identifiers | Additional information |
| --- | --- | --- | --- | --- |
| Software, algorithm | Matlab | Matlab | RRID:SCR_001622 | |
| Software, algorithm | Python | Python | RRID: SCR_008394 | |

For this study, we used two adult male rhesus monkeys (*Macaca mulatta*) that have been extensively trained on the direction discrimination (dots) task. All training, surgery, and experimental procedures were in accordance with the National Institutes of Health Guide for the Care and Use of Laboratory Animals and were approved by the University of Pennsylvania Institutional Animal Care and Use Committee (protocol # 804726).

## Task design and electrophysiology

The behavioral task (*Figure 1A*), general surgical procedure, and data acquisition methods have been described in detail previously (*Ding and Gold, 2010*; *Ding and Gold, 2012b*). Briefly, the monkey was required to report the perceived motion direction of the random-dot stimulus with a saccade at a self-determined time. Trials with different motion coherences (drawn from five levels) and directions were interleaved randomly. The monkey's eye position was monitored with a video-based eye tracker and provided reward/error feedback online based on comparisons between the monkey's eye position

and task-relevant locations. Saccade RT was measured offline with established velocity and acceleration criteria. Neural activity was recorded using glass-coated tungsten electrodes (Alpha-Omega) or polyamide-coated tungsten electrodes (FHC, Inc), using a grid system through a recording chamber with access to the STN. For microstimulation sessions, lower-impedance FHC electrodes were used to record and stimulate at the same sites. Single units were identified by offline spike sorting (Offline Sorter, Plexon, Inc). Electrical microstimulation was delivered using Grass S88 stimulator as a train of negative-leading bipolar current pulses (250 µs pulse duration, 200 Hz) from motion onset to saccade onset. For most sessions, a current intensity of 50 µA was used. In other sessions, we lowered the intensity to ensure that microstimulation did not abolish the monkey's ability to complete the trials. We randomly interleaved trials with and without microstimulation at a 1:1 ratio.

## Localizing the STN

We obtained structural MRI scans using T1-MPRAGE and/or T2-SPACE sequences. We estimated the likely chamber coordinates with access to the STN from these images (and 3D reconstruction using BrainSight from Rogue Research, Inc) and mapped the surrounding areas electrophysiologically. Specifically, we identified several putative landmark regions, including: (1) thalamus, which showed characteristic bursts of activity in a low-firing background while the monkey dozed off; (2) reticular nucleus of the thalamus, where neurons exhibited high baseline firing rates (with bursts sometimes >100 Hz); (3) zona incerta, where neurons exhibited low, tonic baseline firing and briefly paused their activity around saccades (*Ma, 1996*); (4) substantia nigra, pars reticulata, where some neurons showed high baseline firing rates and suppression in activity around visual stimulus or saccade onset (*Hikosaka and Wurtz, 1983*); and (5) substantia nigra, pars compacta, where neurons showed low baseline firing and responded to unexpected reward. Based on a macaque brain atlas (*Saleem and Logothetis, 2007*) and previously reported STN activity patterns (*Matsumura et al., 1992*; *Wichmann et al., 1994b*; *Isoda and Hikosaka, 2008*), we defined STN as the area that: (1) was surrounded by these landmark regions, (2) was separated from them by gaps with minimal activity (white matter), and (3) exhibited irregular firing patterns with occasional short bursts. The baseline firing rate, measured within 50 ms before fixation point onset, had a mean ± SD magnitude of 15.4±12.4 spikes/s in our sample.

## Neural-activity analysis

We measured the firing rates for each neuron and trial condition in running windows (300 ms) aligned to motion and saccade onsets. To visualize the overall activation/suppression, we averaged the firing rates across trial conditions and computed the *z*-scores using a 300 ms window before motion onset as the baseline. To quantitatively measure each neuron's task-related modulation, we performed two multiple linear regressions for each running window, separately for coherence and RT because monkeys' RT strongly depends on coherence on our task:

$$Spike\ count = \beta_0 + \beta_{Choice} \times I_{Choice} + \beta_{Coh-Contra} \times I_{Coh-Contra} + \beta_{Coh-Ipsi} \times I_{Coh-Ipsi} \quad (1)$$

$$Spike\ count = \beta_0 + \beta_{Choice} \times I_{Choice} + \beta_{RT-Contra} \times I_{RT-Contra} + \beta_{RT-Ipsi} \times I_{RT-Ipsi} \quad (2)$$

where,

$$I_{Choice} = \{1\ \text{for contralateral choice}, -1\ \text{for ipsilateral choice}\}$$

$$I_{Coh-Contra} = \{\text{coherence for contralateral choice}, 0\ \text{for ipsilateral choice}\},$$

$$I_{Coh-Ipsi} = \{0\ \text{contralateral choice}, \text{coherence for ipsilateral choice}\},$$

$$I_{RT-Contra} = \{\text{RT for contralateral choice}, 0\ \text{for ipsilateral choice}\},$$

$$I_{RT-Ipsi} = \{0\ \text{for contralateral choice}, \text{RT for ipsilateral choice}\}.$$

Contralateral/ipsilateral choices refer to saccades toward the target contralateral/ipsilateral to the recording sites. Significance of non-zero coefficients was assessed using a *t*-test (criterion: p=0.05).

## Cluster analysis

We converted each neuron's activity into a 30D vector consisting of the average firing rate within three 200 ms windows for all trial conditions (i.e. 2 choices × 5 coherence levels). The windows were selected as early motion viewing (100–300 ms after motion onset), late motion viewing (300–500 ms

after motion onset), and peri-saccade (100 ms before to after saccade onset). The choice identity was designated as either 'preferred' and 'other', based on the relative average activity in the peri-saccade window. Note that this designation was used so that neurons with similar general modulation patterns except for the polarity of their choice selectivity would be grouped together. This designation was not based on any statistical test and did not imply that the peri-saccade activity was reliably choice selective. The average firing rate for each neuron was then z-scored based on baseline rates measured in a 300 ms window ending at motion onset.

We explored multiple method variations using k-means clustering and present results from the variation with the highest stability. These variations included: (1) whether or not the vectors were projected onto 11 principal components that together explained at least 95% of total variance; and (2) calculation of vector distance, including squared Euclidean, cosine, and correlation metrics. We determined the best settings using: (1) the Rand index (**Rand, 1971**), which quantifies the stability of clusters in repeated clustering; (2) silhouette scores, which quantify the quality of grouping and separation between clusters; and (3) visual inspection of clustering results in terms of both cluster distribution in a t-SNE space and average activity of the clusters. To compute the Rand index, we performed 50 runs of clustering, assuming three to nine clusters, for each combination of variations. The Rand index was computed as the fraction of consistent grouping between a pair of units between two clustering runs. For two runs of clustering results, Rand index $= \frac{N_{same-same} + N_{different-different}}{N_{all\ pairs}}$, where $N_{same-same}$ counts the number of neuron pairs that share clusters in both runs, $N_{different-different}$ counts the number of neuron pairs that do not share clusters in either run, and $N_{all\ pairs}$ counts the total number of neuron pairs. To compute the silhouette scores, we chose the best of 100 repetitions of clustering for each combination of variations. For each neuron, silhouette score $= \frac{\max(D_{inter-cluster}, D_{intra-cluster})}{D_{inter-cluster} - D_{intra-cluster}}$, where $D_{inter-cluster}$ is the average distance to the neuron's nearest neighboring cluster, and $D_{intra-cluster}$ is the average distance to other neurons in the same cluster. A positive score implies that, for the given neuron, its activity was more similar to other neurons within the same cluster than those in its nearest neighboring cluster. A negative score implies that the neuron's activity was more similar to those outside its own cluster.

To classify activity recorded at a microstimulation site, we calculated the correlation between its 30D vector and the centroids from random-seeded clustering. The centroid with the highest correlation value determined the cluster identity of the activity.

## Microstimulation-effects analysis

We analyzed microstimulation effects in several ways. To characterize the effects without assumptions about the underlying decision process, we fitted logistic functions to the choice data and linear functions to the RT data. We used three variants of the logistic functions that differed in their use of lapse rates, which measure the probability of errors independent of motion strength:

No Lapse:

$$p\,(contralateral\,choice) = \frac{1}{1+e^{-(Slope_0 + Slope_{estim}) \times (Coh + Bias_0 + Bias_{estim})}} \tag{3}$$

Symmetric Lapse:

$$p\,(contralateral\,choice) = \lambda_0 + \lambda_{estim} + \frac{1 - 2 \times (\lambda_0 + \lambda_{estim})}{1 + e^{-(Slope_0 + Slope_{estim}) \times (Coh + Bias_0 + Bias_{estim})}} \tag{4}$$

Asymmetric Lapse:

$$p\,(contralateral\,choice) = \lambda_{Ipsi0} + \lambda_{Ipsi-estim} + \frac{1 - \lambda_{Ipsi0} - \lambda_{Ipsi-estim} - \lambda_{Contra0} - \lambda_{Contra-estim}}{1 + e^{-(Slope_0 + Slope_{estim}) \times (Coh + Bias_0 + Bias_{estim})}} \tag{5}$$

where Coh is the signed coherence (positive/negative for motion toward the contralateral/ipsilateral choice). Contralateral/ipsilateral choices refer to saccades toward the targets contralateral/ipsilateral to the microstimulation sites, respectively. To assess the significance of the 'estim' terms, we used bootstrap methods. Specifically, we generated 200 sets of data by shuffling the microstimulation status of trials within each session. We fitted these artificial data using the same logistic functions to estimate null distributions for each parameter and performed a one-tailed test to determine if the actual fit value exceeded chance (criterion, $p < 0.05$).

We fitted linear functions to the RT data, separately for the two choices:

$$RT = Offest_0 + Offset_{estim} + \left(Slope_0 + Slope_{estim}\right) \times Coh_{unsigned} \tag{6}$$

We assessed significance using $t$-tests (criterion, p<0.05).

To infer microstimulation effects on decision-related computations, we fitted DDM to choice and RT data simultaneously. We used DDM variants with collapsing bounds (DDM; *Figure 7A*), following previously established procedures (*Fan et al., 2018*; *Doi et al., 2020*). Briefly, the DDM assumes that motion evidence is accumulated over time into a DV, which is compared to two collapsing choice bounds. A choice is made when the DV crosses either bound, such that the time of crossing determines the decision time and the identity of the bound determines the choice identity. The model has eight basic parameters (presented here in six groups): (1) $a$, the maximal bound height; (2) $B\_collapse$ and $B\_t$, the decay speed and onset specifying the time course of the bound 'collapse', respectively; (3) $k$, a scale factor governing the rate of evidence accumulation; (4) $me$, an offset specifying a bias in the rate of evidence accumulation; (5) $z$, an offset specifying a bias in the DV, or equivalently, asymmetric offsets of equal magnitude for the two choice bounds; and (6) $t0_{contra}$ and $t0_{ipsi}$, non-decision times for the two choices that capture RT components that do not depend on evidence accumulation (e.g. visual latency and motor delay).

We used eight variants of DDM. In the Full model, all eight parameters were allowed to change with microstimulation. In the None model, all eight parameters did not change with microstimulation. In six reduced models (NoA, NoCollapse, NoK, NoME, NoZ, NoT), the corresponding group of parameters (specified above) were fixed while the other parameters were allowed to change with microstimulation. We fitted each model using the maximum a posteriori estimate method and previously established prior distributions (*Wiecki et al., 2013*). We performed five runs for each fit and used the best run (highest likelihood) for analyses here. We used the AIC for model selection. We considered an AIC difference >3 to indicate that the smaller-AIC model significantly outperformed the larger-AIC model. For a given sessions, if the Full model outperformed a reduced model and the None model, we considered that session to show significant microstimulation effect(s) on the corresponding model parameter(s). For example, we considered STN microstimulation to induce significant changes in $k$ if the Full model outperformed both None and NoK models for a given session.

## Acknowledgements

We thank Jean Zweigle for outstanding animal care and training, Lowell Thompson and Kara McGaughey for comments on the manuscript, and Michael Suplick for machine shop support (NIH National Eye Institute Core Grant P30 EY001583). This work was supported by NIH National Eye Institute (R01-EY022411; LD and JIG).

## Additional information

### Competing interests

Joshua I Gold: Senior editor, *eLife*. The other authors declare that no competing interests exist.

### Funding

| Funder | Grant reference number | Author |
| --- | --- | --- |
| National Eye Institute | R01-EY022411 | Joshua I Gold<br>Long Ding |
| National Eye Institute | Core Grant P30 EY001583 | Joshua I Gold<br>Long Ding |

The funders had no role in study design, data collection and interpretation, or the decision to submit the work for publication.

### Author contributions

Kathryn Branam, Data curation, Investigation, Methodology, Writing – review and editing; Joshua I Gold, Conceptualization, Formal analysis, Funding acquisition, Visualization, Methodology, Writing

– review and editing; Long Ding, Conceptualization, Resources, Data curation, Formal analysis, Supervision, Funding acquisition, Validation, Investigation, Visualization, Methodology, Writing - original draft, Project administration, Writing – review and editing

**Author ORCIDs**
Joshua I Gold https://orcid.org/0000-0002-6018-0483
Long Ding https://orcid.org/0000-0002-1716-3848

**Ethics**
All training, surgery, and experimental procedures were in accordance with the National Institutes of Health Guide for the Care and Use of Laboratory Animals and were approved by the University of Pennsylvania Institutional Animal Care and Use Committee (protocol # 804726).

Reviewer #2 (Public review): https://doi.org/10.7554/eLife.98345.3.sa1
Reviewer #3 (Public review): https://doi.org/10.7554/eLife.98345.3.sa2
Author response https://doi.org/10.7554/eLife.98345.3.sa3

## Additional files

**Supplementary files**
• Supplementary file 1. Indices of motivational state did not correlate with microstimulation effects. p-Values are raw values from Pearson correlation, not corrected for multiple testing.
• MDAR checklist

**Data availability**
Data and code are available at Open Science Framework (https://osf.io/z3596/).

The following dataset was generated:

| Author(s) | Year | Dataset title | Dataset URL | Database and Identifier |
| --- | --- | --- | --- | --- |
| Branam K, Gold JI, Ding L | 2024 | STN | https://osf.io/z3596/ | Open Science Framework, z3596 |

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
