## [Editor Report · eLife assessment]

The **fundamental** study by Ding and colleagues identifies subpopulations of neurons recorded in the monkey subthalamic nucleus (STN) with distinct activity profiles and causal contributions during perceptual decision-making. The combination of neuronal recording, microstimulation, and computational methods provides **convincing** evidence for a heterogenous neural population that could support multifaceted roles in decision formation. This study should be of wide interest to computational and experimental neuroscientists interested in cognitive function.

---

## [Referee Report · Reviewer #2 (Public review)]

This study uses single-unit recordings in the monkey STN to examine the evidence for three theoretical models that propose distinct roles for the STN in perceptual decision-making. Importantly, the proposed functional roles are predictive of unique patterns of neural activity. Using k-means clustering with seeds informed by each model's predictions, the current study identified three neural clusters with activity dynamics that resembled those predicted by the described theoretical models. The authors are thorough and transparent in reporting the analyses used to validate the clustering procedure and the stability of the clustering results. To further establish a causal role for the STN in decision-making, the researchers applied microsimulation to the STN and found effects on response times, choice preferences, and latent decision parameters estimated with a drift-diffusion model. Overall, the study provides strong evidence for a functionally diverse population of STN neurons that could indeed support multiple roles involved in perceptual decision-making. The manuscript would benefit from stronger evidence linking each neural cluster to specific decision roles in order to strengthen the overall conclusions.

The interpretation of the results, and specifically, the degree to which the identified clusters support each model, is largely dependent on whether the artificial vectors used as model-based clustering seeds adequately capture the expected behavior under each theoretical model. The manuscript would benefit from providing further justification for the specific model predictions summarized in Figure 1B. Further, although each cluster's activity can be described in the context of the discussed models, these same neural dynamics could also reflect other processes not specific to the models. That is, while a model attributing the STN's role to assessing evidence accumulation may predict a ramping up of neural activity, activity ramping is not a selective correlate of evidence accumulation and could be indicative of a number of processes, e.g., uncertainty, the passage of time, etc.. This lack of specificity makes it challenging to infer the functional relevance of cluster activity and should be acknowledged in the discussion.

Additionally, although the effects of STN microstimulation on behavior provide important causal evidence linking the STN to decision processes, the stimulation results are highly variable and difficult to interpret. The authors provide a reasonable explanation for the variability, showing that neurons from unique clusters are anatomically intermingled such that stimulation likely affects neurons across several clusters. It is worth noting, however, that a substantial body of literature suggests that neural populations in the STN are topographically organized in a manner that is crucial for its role in action selection, providing "channels" that guide action execution. The authors should comment on how the current results, indicative of little anatomical clustering amongst the functional clusters, relates to other reports showing topographical organization.

Overall, the association between the identified clusters and the function ascribed to the STN by each of the models is largely descriptive and should be interpreted accordingly. For example, Figure 3 is referenced when describing which cluster activity is choice/coherence dependent, yet it is unclear what specific criteria and measures are being used to determine whether activity is choice/coherence "dependent." Visually, coherence activity seems to largely overlap in panel B (top row). Is there a statistically significant distinction between low and high coherence in this plot? The interpretation of these plots and the methods used to determine choice/coherence "dependence" needs further explanation.

In general, the association between cluster activity and each model could be more directly tested. At least two of the models assume coordination with other brain regions. Does the current dataset include recordings from any of these regions (e.g., mPFC or GPe) that could be used to bolster claims about the functional relevance of specific subpopulations? For example, one would expect coordinated activity between neural activity in mPFC and Cluster 2 according to the Ratcliff and Frank model. Additionally, the reported drift-diffusion model (DDM) results are difficult to interpret as microsimulation appears to have broad and varied effects across almost all the DDM model parameters. The DDM framework could, however, be used to more specifically test the relationships between each neural cluster and specific decision functions described in each model. Several studies have successfully shown that neural activity tracks specific latent decision parameters estimated by the DDM by including neural activity as a predictor in the model. Using this approach, the current study could examine whether each cluster's activity is predictive of specific decision parameters (e.g., evidence accumulation, decision thresholds, etc.). For example, according to the Ratcliff and Frank model, activity in cluster 2 might track decisions thresholds.

Review of revision

The authors have sufficiently addressed the concerns raised in the initial reviews and have revised their manuscript accordingly. We commend the authors for these efforts and feel that the revisions have strengthened the major claims of the manuscript.

---

## [Referee Report · Reviewer #3 (Public review)]

Summary:

The authors provide compelling evidence for the causal role of the subthalamic nucleus (STN) in perceptual decision-making. By recording from a large number of STN neurons and using microstimulation, they demonstrate the STN's involvement in setting decision bounds, scaling evidence accumulation, and modulating non-decision time.

Strengths:

The study tested three hypotheses about the STN's function and identified distinct STN subpopulations whose activity patterns support predictions from previous computational models. The experiments are well-designed, the analyses are rigorous, and the results significantly advance our understanding of the STN's multi-faceted role in decision formation.

Weaknesses:

While the study provides valuable insights into the STN's role in decision-making, there are a few areas that could be improved. First, the interpretation of the neural subpopulations' activity patterns in relation to the computational models should be clarified, as the observed patterns may not directly correspond to the specific signals predicted by the models. Second, a neural population model could be employed to better understand how the STN population jointly contributes to decision-making dynamics.

---

## [Author Response]

The following is the authors’ response to the original reviews.

**Review #1:**
(1) It would be helpful to explain the criteria for choosing a given number of clusters and for accepting the final clustering solution more clearly. The quantitative results (silhouette plots, Rand index) in Supplementary Figure 2 should perhaps be included in the main figure to justify the parameter choices and acceptance of specific clustering solutions.

We revised the text and added labels to the original Supplementary Figure 2 (now main Figure 4) to clarify how we arrived at the best settings for random-seed clustering.

(2) It would be helpful to show how the activity profiles in Figure 3 would look like for 3 or 5 (or 6) clusters, to give the reader an impression of how activity profiles recovered using different numbers of clusters would differ.

We added a new figure (Supplementary Figure 4) that shows 5- and 6-cluster results. Note that the same three subpopulations in Figure 3 were reliably identified as distinct clusters even with alternative settings, corroborating the results in the tSNE space (Supplementary Figure 3).

(3) The authors attempt to link the microstimulation effects to the presence of functional neuron clusters at the stimulation site. How can you rule out that there were other, session-specific factors (e.g., related to the animal's motivation) that affected both neuronal activity and behavior? For example, could you incorporate aspects of the monkey's baseline performance (mean reaction time, fixation breaks, error trials) into the analysis?

We tested the potential influences of monkeys’ motivational states on our observations using two sets of analysis. First, we examined whether motivational state modulated the likelihood of observing a specific type of neural activity in STN. We focused on three measurements of motivational states: the rate of fixation break, the overall error rate, and mean RT. We found that none of these measurements differed significantly among sessions when we encountered different subpopulations (new Supplemental Figure 7), suggesting that motivational state alone cannot explain the differences in activity patterns of the four subpopulations.

Second, we examined how motivational state may be reflected in the microstimulation results. To clarify, because we interleaved trials with and without microstimulation, the microstimulation effects cannot be solely explained by session-specific factors. However, it is possible that motivational state can modulate the magnitude of microstimulation effects. We performed correlation analysis between microstimulation effects (difference in each fitted DDM parameter between trials with and without microstimulation) and motivational state (fixation break, error rate, mean RT on trials without microstimulation). We did not find significant correlation for any combination (Supplemental Table 1). These results suggest that the motivational state of the monkey had little influence on our recording and microstimulation results. However, because our monkeys operated within a narrow range of strong engagement on the task, we cannot rule out the possibility that STN activity or microstimulation effects could change significantly if the monkeys were not as engaged. We have added these results in a new section titled “Heterogeneous activity patterns and microstimulation effects cannot be explained by variations in motivational state”.

(4) Line 84: What was the rationale for not including both coherence and reaction time in one multiple regression model?

On the task we used, RT depends strongly on coherence in a nonlinear fashion (e.g., example behavior in now Figure 5). We thus performed regressions using coherence and RT separately. We revised the text in Methods to clarify our rationale (lines 470-473):

“To quantitatively measure each neuron’s task-related modulation, we performed two multiple linear regressions for each running window, separately for coherence and RT because monkeys’ RT strongly depends on coherence on our task:”

**Review #2**:The interpretation of the results, and specifically, the degree to which the identified clusters support each model, is largely dependent on whether the artificial vectors used as model-based clustering seeds adequately capture the expected behavior under each theoretical model. The manuscript would benefit from providing further justification for the specific model predictions summarized in Figure 1B.

We added information on the original figure/equations that were the basis of the artificial vectors we constructed for clustering analysis and their abbreviated summary in Figure 1B (first paragraph in section “STN subpopulations can support previously theorized functions”). These vectors were meant to capture prominent features of the predicted activity patterns, in the forms of choice, time, and motion strength dependencies. We also emphasize that we obtained very similar results using random clustering seeds.

Further, although each cluster's activity can be described in the context of the discussed models, these same neural dynamics could also reflect other processes not specific to the models. That is, while a model attributing the STN's role to assessing evidence accumulation may predict a ramping up of neural activity, activity ramping is not a selective correlate of evidence accumulation and could be indicative of a number of processes, e.g., uncertainty, the passage of time, etc. This lack of specificity makes it challenging to infer the functional relevance of cluster activity and should be acknowledged in the discussion.

We thank the reviewer for pointing out the alternative interpretation of these modulation patterns. We have added this caveat in the Discussion (lines 398-401): “It is also possible that the ramping activity reflects alternative roles for the STN in the evaluation of the decision process, the tracking of elapsed time, or both. How these possible roles relate to those of caudate neurons awaits further investigation (Fan et al., 2024)”.

Additionally, although the effects of STN microstimulation on behavior provide important causal evidence linking the STN to decision processes, the stimulation results are highly variable and difficult to interpret. The authors provide a reasonable explanation for the variability, showing that neurons from unique clusters are anatomically intermingled such that stimulation likely affects neurons across several clusters. It is worth noting, however, that a substantial body of literature suggests that neural populations in the STN are topographically organized in a manner that is crucial for its role in action selection, providing "channels" that guide action execution. The authors should comment on how the current results, indicative of little anatomical clustering amongst the functional clusters, relate to other reports showing topographical organization.

We thank the reviewer for raising this important point. We have added the following text in the Discussion:

“The intermingled subpopulations may appear at odds with the conventional idea of topography in how the STN is organized. For example, the “tripartite model” suggests that STN is segregated by motor, associative, and limbic functions (Parent and Hazrati, 1995); afferents from motor cortices and neurons related to different types of movements are largely somatotopically organized in the STN (DeLong et al., 1985; Nambu et al., 1996); and certain molecular markers are expressed in an orderly pattern in the STN (reviewed in Prasad and Wallén-Mackenzie, 2024). Because we focused on STN neurons that were responsive on a single oculomotor decision task, our sampling was likely biased toward STN subdivisions related to associative function and oculomotor movements. As such, our results do not preclude the presence of topography at a larger scale. Rather, our results underscore the importance of activity patternbased analysis, in addition to anatomy-based analysis, for understanding the functional organization of the STN.”

Figure 3 is referenced when describing which cluster activity is choice/coherence dependent, yet it is unclear what specific criteria and measures are being used to determine whether activity is choice/coherence "dependent." Visually, coherence activity seems to largely overlap in panel B (top row). Is there a statistically significant distinction between low and high coherence in this plot? The interpretation of these plots and the methods used to determine choice/coherence "dependence" needs further explanation.

We added a new figure (Sup Figure 3) that shows the summary of choice and coherence modulation, based on multiple linear regression analysis, for each subpopulation separately. We also updated the description of these activity patterns in Results (lines 122-130):

In general, the association between cluster activity and each model could be more directly tested. At least two of the models assume coordination with other brain regions. Does the current dataset include recordings from any of these regions (e.g., mPFC or GPe) that could be used to bolster claims about the functional relevance of specific subpopulations? For example, one would expect coordinated activity between neural activity in mPFC and Cluster 2 according to the Ratcliff and Frank model.

We agree completely that simultaneous recordings of STN and its afferent/efferent regions (such as mPFC, GPe, SNr, and GPi) would provide valuable insights into the specific roles of STN and the basal ganglia as a whole. Such recordings are outside the scope of the current study but are in our future plans.

Additionally, the reported drift-diffusion model (DDM) results are difficult to interpret as microstimulation appears to have broad and varied effects across almost all the DDM model parameters. The DDM framework could, however, be used to more specifically test the relationships between each neural cluster and specific decision functions described in each model. Several studies have successfully shown that neural activity tracks specific latent decision parameters estimated by the DDM by including neural activity as a predictor in the model. Using this approach, the current study could examine whether each cluster's activity is predictive of specific decision parameters (e.g., evidence accumulation, decision thresholds, etc.). For example, according to the Ratcliff and Frank model, activity in cluster 2 might track decision thresholds.

We thank the reviewer for the suggested analysis. Because including the neural activity in the model substantially increases model fitting time, we performed a preliminary round of model fitting for 15 neurons (5 neurons closest to each of the cluster centroids). For each neuron, we measured the average firing rates in three windows: (1) a 350 ms window starting from dots onset (“Dots”), (2) a 350 ms window ending at saccade onset (“Presac”), and (3) a variable window starting from dots onset and ending at 100 ms before saccade onset (“Fullview”). For each window, the firing rates were z-scored across trials. We incorporated the firing rates into two model types. In the “DV” type, the firing rates were assumed to influence three DDM parameters related to evidence accumulation: k, me, and z. In the “Bound” type, the firing rates were assumed to influence three DDM parameters related to decision bound: a, B_alpha, and B_d. In total, we fitted six combinations of firing rates and model types to each neuron. For comparison, we also fitted the standard model without incorporating firing rates.

As shown in Author response image 1, firing rates of single STN neurons had minimal contributions to the fits. With the exception of one neuron, AIC values were greater for model variants including firing rates than the standard model (Author response image 1A), indicating that including firing rate did not improve the fits. For all neurons, the actual fitted coefficients for firing rates were several degrees of magnitude smaller than the corresponding DDM parameter (Author response image 1B; note the range of y axis), indicating that the trial-by-trial variation in firing rate had little influence on the evidence accumulation- or decision bound-related parameters. Based on these preliminary fitting results, we believe that a single STN neuron does not have strong enough influence on the overall evidence accumulation or decision bound to be detected with the model fitting method. We therefore did not expand the fitting analysis to all neurons.

**Author response image 1. sa3fig1:** Firing rates of a single STN neuron did not substantially influence decision-related DDM parameters. (**A**) Differences in AIC between DDM variants that included firing rate-dependent terms and the standard DDM. Red dahsed line: difference = -3. Each column represents results from one unit. (**B**) Fitted coefficients for firing rate-related terms were near zero. Note the range of y axis. Values for the top and bottomw panels were obtained from "DV"- and "Bound"-type models, respectively. See text for more details.

We emphasize, however, that the apparent negative results do not necessarily argue against a causal role of the STN in decision making, rather, these results more likely reflect the methodological limitation: because we used a single task context, the monkeys’ natural trial-by- trial variations in the DDM components may be too small. A better design would be to manipulate task contexts to induce larger changes in evidence accumulation or decision bounds and then test for a correlation between single-neuron firing rates and these changes. We are currently using such a design in a follow-up study.

The table in Figure 1B nicely outlines the specific neural predictions for each theoretical model but it would help guide the reader if the heading for each column also included a few summary words to remind the reader of the crux of each theory, e.g. "Ratcliff+Frank 2012 (adjusted decision-bounds)"

We thank the reviewer for this suggestion. We considered implementing this but eventually decided not to add more headings to the column, because the predicted STN functions of the three models cannot all be succinctly summarized. We thus prefer to include more detailed descriptions in the main text, instead of in the figure.

The authors frequently refer to contralateral vs. ipsilateral decisions but never explicitly state what this refers to, i.e. contralateral relative to what (visual field, target direction, recording site, etc.)? The reader can eventually deduce that this means contralateral to the recording site but this should be explicitly stated for clarity.

We added in Methods:

Line 483: “Contralateral/ipsilateral choices refer to saccades toward the targets contralateral/ipsilateral to the recording sites, respectively.”

Line 535: Contralateral/ipsilateral choices refer to saccades toward the targets contralateral/ipsilateral to the microstimulation sites, respectively.”

Again, for clarity, it would be helpful to explicitly define what the authors mean by "sensitive to choice" when referring to Figure 1B as this could be interpreted to mean left/right or ipsilateral/contralateral.

In the context of Figure 1B, “sensitive to choice” means showing different responses for the two choices in our 2AFC task, regardless of the task geometry. We added explanation in the figure caption.

Color bar labels would be helpful to include in all figures that include plots with color bars.

We apologize for omitting the labels. They are added to Figure 2B and C, Supplemental Fig. 1.

The authors should briefly note what a "lapse term" is when describing the logistic function results.

We revised the text in Results (lines 184-186) and Methods (line 527) to clarify that lapse terms were used to capture errors independent of motion strength.

Are the 3 example sessions in Figure 4 stimulating the same STN site and/or the same monkey? This information should be noted in the caption or main text.

We revised the caption: “A-C, Monkey’s choice (top) and RT (bottom) performance for trials with (red) and without (black) microstimulation for three example sessions (A,B: two sites in monkey C; C: monkey F).”

Figure 3B the authors note that "the last cluster shows little task-related modulation" - what criteria are they using to make this conclusion? By eye, the last cluster and cluster 1 seem to show a similar degree of modulation when locked to motion onset.

We added a new figure (Suppl Figure 2) that shows the summary of choice and coherence modulation, based on multiple linear regression analysis, for each subpopulation separately.

**Reviewer #3:**

We have grouped the reviewer’s public and specific comments by content.

First, the interpretation of the neural subpopulations' activity patterns in relation to the computational models should be clarified, as the observed patterns may not directly correspond to the specific signals predicted by the models. The authors claim that the first subpopulation of STN neurons reflects the normalization signal predicted by the model of Bogacz and Gurney (2007). However, the observed activity patterns only show choice- and coherence-dependent activity, which may represent the input to the normalization computation rather than its output. The authors should clarify this point and discuss the limitations of their interpretation.

We agree with the reviewer that the choice- and coherence-dependent activity pattern does not sufficiently indicate a normalization computation. We interpreted such activity as satisfying a necessary condition for, and therefore consistent with, the theoretical model proposed by Bogacz and Gurney. We have reviewed the text to ensure that we never made the claim that the first subpopulation mediates the normalization.

Second, the authors could consider using a supervised learning method to more explicitly model the pattern correlations between the three profiles. The authors used k-means clustering to identify STN subpopulations. Given the clear distinction between the three types of neural firing patterns, a supervised learning method (e.g., a generalized linear model) could be used as a more explicit encoding model to account for the pattern correlations between the three profiles.

We used two approaches to examine the different response profiles. The “random-seed” approach used non-supervised clustering to probe the functional organization of STN neurons, with no a priori assumption about how many subpopulations may be present. The “model-seed” approach is similar in spirit to what the reviewer suggested: we defined artificial vectors, akin to regressors in a generalized linear model, that showed key modulation features as predicted by previous theoretical models. We then projected the neurons’ activity profiles onto these vectors, akin to performing a regression analysis.

Third, a neural population model could be employed to better understand how the STN population jointly contributes to decision-making dynamics. The single-neuron encoding analysis reveals mixed effects from multiple decision-related functions. To better understand how the STN population jointly contributes to the decision-making process, the authors could consider using a neural population model (e.g., Wang et al., 2023) to quantify the population dynamics.

We agree with the reviewer that a neural population model would be helpful for testing our understanding of the roles of STN. However, we believe that this is premature at the moment because we have no knowledge about how these different subpopulations interact with each other within STN, nor how they interact with other basal ganglia nuclei. We hope our results provide a foundation for future experiments that can provide more specific insights in the roles of each subpopulation, which can then be tested in a neural population model as the reviewer suggested.

Finally, the added value of the microstimulation experiments should be more directly addressed in the Results section, as the changes in firing patterns compared to the original patterns are not clearly evident. The microstimulation results (Figure 7A) do not show significant changes in firing patterns compared to the original patterns (Figure 3B). As microstimulation is used to identify the hypothetical role of the STN beyond the correlational analysis, the authors should more directly address the added value of these experiments in the Results section.

We apologize for the confusion. The average firing rates at the top of original Figure 7A (now Figure 8A) were obtained in recordings just before microstimulation, to document which neuron subpopulation was near the stimulation electrode. We were not able to obtain recordings from the same neurons during microstimulation.

The ordering of the three hypotheses in the Introduction (1) adjusting decision bounds, (2) computing a normalization signal, (3) implementing a nonlinear computation to improve decision bound adjustment, is inconsistent with the order in which they are addressed in the Results section (2, 1, 3). To improve clarity and readability, the authors should consider presenting the hypotheses and their corresponding results in a consistent order throughout the manuscript.

We thank the reviewer for this suggestion. We have reordered the text in Introduction to be consistent.